# Machine Learning Meets Cancer

**DOI:** 10.3390/cancers16061100

**Published:** 2024-03-08

**Authors:** Elena V. Varlamova, Maria A. Butakova, Vlada V. Semyonova, Sergey A. Soldatov, Artem V. Poltavskiy, Oleg I. Kit, Alexander V. Soldatov

**Affiliations:** 1The Smart Materials Research Institute, Southern Federal University, 178/24 Sladkova Str., 344090 Rostov-on-Don, Russia; mbutakova@sfedu.ru (M.A.B.); sergeysoldatov@sfedu.ru (S.A.S.); poltavsky@sfedu.ru (A.V.P.); soldatov@sfedu.ru (A.V.S.); 2Faculty of Computer Science, HSE University, 20 Myasnitskaya Str., 101000 Moscow, Russia; vvsemyonova_1@edu.hse.ru; 3National Medical Research Centre for Oncology, 63 14 Liniya Str., 344037 Rostov-on-Don, Russia; onko-sekretar@mail.ru

**Keywords:** artificial intelligence, machine learning, oncology, radiomics, PET/CT

## Abstract

**Simple Summary:**

This review examines the latest technologies using machine learning (ML) methods, including the use of convolutional neural networks, decision trees, and generative adversarial networks to solve problems concerning cancer recognition, planning optimal treatment strategies, as well as predicting the likelihood of patient survival. The authors also discuss the prospects of using machine learning technologies in medicine. In addition, the authors emphasize the need to address issues such as the anonymization of data received from patients.

**Abstract:**

The role of machine learning (a part of artificial intelligence—AI) in the diagnosis and treatment of various types of oncology is steadily increasing. It is expected that the use of AI in oncology will speed up both diagnostic and treatment planning processes. This review describes recent applications of machine learning in oncology, including medical image analysis, treatment planning, patient survival prognosis, and the synthesis of drugs at the point of care. The fast and reliable analysis of medical images is of great importance in the case of fast-flowing forms of cancer. The introduction of ML for the analysis of constantly growing volumes of big data makes it possible to improve the quality of prescribed treatment and patient care. Thus, ML is expected to become an essential technology for medical specialists. The ML model has already improved prognostic prediction for patients compared to traditional staging algorithms. The direct synthesis of the necessary medical substances (small molecule mixtures) at the point of care could also seriously benefit from the application of ML. We further review the main trends in the use of artificial intelligence-based technologies in modern oncology. This review demonstrates the future prospects of using ML tools to make progress in cancer research, as well as in other areas of medicine. Despite growing interest in the use of modern computer technologies in medical practice, a number of unresolved ethical and legal problems remain. In this review, we also discuss the most relevant issues among them.

## 1. Introduction

Cancer is one of the most common diseases in the world. In 2022, 1,930,000 cases of this disease were detected worldwide. Lung cancer is the leader in regard to the number of deaths. Using computer-aided diagnosis (CAD) can significantly speed up the process of cancer diagnosis. In order to use medical images as input data for CAD, it is necessary to find a way to effectively identify features in images. Using machine learning in CAD systems can improve the accuracy of histopathological diagnoses. Over the past 5 years, AI has demonstrated unprecedented progress in the field of computer vision (CV). Methods of extracting dependencies from data, implemented with the help of AI, represent great potential for cancer diagnostics. AI technologies are beginning to be widely used in screening, diagnosis, predicting future outcomes, disease monitoring, treatment planning, and clinical oncology research in general. An important novel direction is the development of computer-driven systems for personalized medicine, namely the individual synthesis of the necessary medical substances directly at the bedside. Thus, this approach could be of great importance for the future success of medicine.

## 2. Diagnostics

Radiomics is the subdiscipline of AI dealing with the computation, identification, and extraction of image features, as well as the generation of prognostic or predictive mathematical models. It is a novel, developing, supplementary source of diagnostic information used for analyzing clinical data based on ML [1]. It is a non-invasive method. In silico research needs data to make a breakthrough for this clinical approach. It is possible to identify various parameter correlations. It only requires the data to be available for a representative sample of patients. This is the reason why the benefits of using AI are extensive.

One of the problems in radiation oncology today is the uncertainty in creating medical diagnostic image descriptions, which directly affect the model for predicting treatment and patient care. This is often expressed by the large variability in the definition of images, represented by the structural marks made by the doctor. In a prior study [2], W.C. Sleeman et al. proposed an approach to change the description of images according to the American Association of Physicists in Medicine’s (AAPM) Task Group 263 (TG- 263). Scientists selected datasets with diagnostic images for the first 709 patients with lung cancer and 752 patients with prostate cancer from 40 medical centers. Training datasets were collected as part of the Radiation Oncology Quality Surveillance Program (VAROQS). A comparative analysis was carried out on the Apache Spark platform using machine learning algorithms, such as Naive Bayes (NB), Random Forest (RF), Gradient Boost Tree (GBT), Multilayer Perceptron (MLP), and Support Vector Machine (SVM) techniques. The authors constructed signs of a bony anatomy that were used as additional vectors for the comparative analysis. The optimization of the learning process was performed by reducing the dimension of these features using the singular value decomposition (SVD), which gave an error within 0.1% at a higher speed, in contrast to the traditional principal component analysis (PCA). The data from 50 lung cancer patients and 50 prostate cancer patients, provided by Virginia Commonwealth University (VCU), were used as test datasets for validating the pre-trained models. The best result was demonstrated by the RF algorithm at 98.77 for lung cancer and 95.06 for prostate cancer. In general, using the prepared data, all the methods showed accuracy at the level of 90%, except for NB, and using the clinical data, the results was comparable to the Virginia Commonwealth University (VCU) external test dataset, and here MLP showed itself to be the best of all. The presence of signs of a bony anatomy in all cases increased the accuracy of the measurements, so even when using clinical kits, the results were 95% for lung cancer and 91% for prostate cancer.

An indicator of normal or pathogenic biological processes in the body or reactions to therapeutic intervention is a biomarker. F. Klauschen et al., in a prior study [3], describe how a qualitative analysis of a new intracranial formation plays a crucial role in diagnosis and can determine whether a confirmatory biopsy or resection should be performed in clinical practice. Their research in the field of image analysis is aimed at extracting information from a set of image data for further creation of biomarkers, the schematic diagram of which is shown in Figure 1.

There are several phases of radiomics: image preprocessing, feature evaluation, feature selection, classification, and evaluation. An important part of the study is preprocessing: correction of intensity, unevenness, as well as noise reduction. The ML methods can be successfully used to solve the problems described above. This review describes the studies aimed at visualizing biomarkers for monitoring human responses to treatment, as well as for diagnosis and prognosis. The use of the imaging method would make it possible to distinguish patients with true progression from patients with pseudo-progression. This would make it possible to change the treatment strategy and prevent ineffective treatment. As a result of the selection of signs, 600,000 positron emission tomography (PET) clusters based on three classes were obtained, and the accuracy of the diagnosis of pseudo-progression and tumor progression was 90%. It was proved that in several types of cancer, the number of tumor-infiltrating lymphocytes (TILs), along with other biomarkers, is a marker of the response to immune checkpoint therapy. In their paper, A. Klauschen et al. discuss automated approaches to calculate the number of TILs. Both classical approaches and non-traditional approaches were considered. Explicable ML allows medical specialists to interpret the result for patients. The ability to scan the whole slide images (WSIs) and the availability of solutions that can process the WSIs are attracting much attention to digital pathology. Breast cancer is an example where TIL scoring is important. The article discusses various predictors of the body’s response to therapy. The authors consider segmentation and direct classification methods. Classical segmentation methods require small amounts of input data. Among them, threshold segmentation, the watershed method, and segmentation based on morphology are considered. The main issue in applying the direct approach is that the typical size of the input image for the deepest neural networks (NNs) is 300 × 300 pixels. Currently, the WSIs have dimensions up to 10,000 × 10,000 pixels. The authors discuss other approaches in which restrictive frames are displayed around the cells as an alternative to segmentation. Examples of such methods include Fast-R-CNN, Faster R-CNN, and others. In such methods, a sliding window is used, which moves around the image and transmits location information if the window falls into a certain cell. The result and its reliability are not always easy for a specialist to interpret. There are several approaches that are used to explain the prediction result, such as Guided Backprop, sensitivity analysis, integrated gradients, and DeepLIFT. These methods work in different ways. For example, a gradient-based method shows which regions make the greatest contribution to the prediction. The authors note that in order to use ML approaches in cancer diagnosis everywhere, it is necessary to develop clear criteria and evaluation standards using benchmark datasets.

Big data phenomena give new opportunities in radiation oncology as well. However, the problem of effectively processing big data retrieved from multiple sources and different locations for radiation oncology clinics remains unsolved. J. Kazmierska et al., in their work [4], tried to identify the barriers and limitations of big data adoption in the clinical data science community in radiation oncology. Multisource big data in radiation oncology clinics arise from constantly increasing data flows, including electronic health records, patients’ laboratory measures, output data, medical images (PET, computed tomography (CT), cone-beam computed tomography (CBCT), magnetic resonance imaging (MRI)), and the features extracted from images. The authors noticed that the clinicians themselves were not able to process such huge datasets without clear rules, and the AI approaches coupled with AI/ML could be really beneficial for scientists.

In [5], R. Seifert et al. discuss in detail the application of AI in various fields of nuclear medicine imaging and the directions of advanced scientific developments. The authors provide examples of applying AI for image analysis, pre- and post-processing of images, as well as for predicting the success of treatment and psychotherapy.

Although the use of AI in nuclear medicine imaging is at an early stage of its development, there is great scientific potential for the use of imaging systems in clinical practice. There are several ways to use image analysis as a part of AI for the segmentation of tumor PETs and for the quantification of a whole-body tumor volume. Currently AI is being used to improve fade correction of PET images, image capture and recovery, automated disease classification, and automated metastatic detection. The authors of the study emphasize three approaches to using image analysis in nuclear medicine. First, ML is employed for the specification of body subregions. Second, AI is employed to quantify the whole-body tumor volume in PET acquisitions. Finally, AI for pathological accumulation segmentation is commonly used in a variety of cancer research, like prostate-specific membrane antigen (PSMA)-PET, bone scintigraphy, or fluorodeoxyglucose (FDG-PET). ML imagining is carried out in nuclear cardiology to determine whether surgical intervention is necessary for the patient. PET and SPECT usually combine imaging scans with CT. The study analyzed the data from 713 patients who underwent myocardial perfusion scintigraphy and received cardiac catheterization. Based on these results, the data parameters that represented information gain were selected from 33 total parameters and a prediction model was then trained. The algorithm showed the results comparable or superior to human operators in predicting the need for surgical intervention. The application of ML makes it possible to integrate many variables that a person is not able to supervise. AI and ML are also actively used in dealing with benign diseases. In addition to classification, AI effectively correlates the dependence of various clinical factors with treatment options.

Individual processing can also be performed by AI, this is especially important in time-consuming processes such as complete image reconstruction. This leads to a decrease in the number of indicators used and prevents the need for simultaneous acquisition of CT, which results in a decrease in radiation exposure. The data collection time is also reduced, which increases the number of studies.

One of the main promising directions of the ML development for cancer diagnostics is the separation of a metastatic tumor in the body, taking into account various parameters of metastases and tumors, the accurate determination of changes in the volume of each metastasis, the study of the tumor for lesions in vivo, the analysis and processing of dynamic PET data, and automatic image noise reduction.

The confidentiality of patient data while organizing the general access to medical diagnostic data for solving specialized problems is an important issue for the healthcare sector in terms of progress and innovation development. The study in [6] presents a distributed structure of Personal Health Train (PHT) based on the FAIR data model (Findable, Accessible, Interoperable, Reusable), which allows ML and data mining in a decentralized manner while maintaining patient privacy. T.M. Deist et al. used the total data coverage at eight medical organizations in five countries, which made it possible to provide information on 23,203 clinical cases. The authors trained the ML Logistic Regression (LR) model on 14,810 patients (treated from 1978 to 2011) and confirmed it on 8393 patients treated from 2012 to 2015. As a result, a comparative analysis has been conducted with an alternative platform for distributed data analysis and ML DataSHIELD. The possible weak points of PHT are also highlighted.

Modern clinical diagnostic studies in medical institutions generate a large amount of heterogeneous data. Precision medicine methods such as next-generation sequencing involve evaluating tens of thousands of parameters when diagnosing one patient. In [7], J.-E. Bibault et al. provided an evidence base for the effectiveness of using Big Data technology for the digital support of modern radiation oncology. A summary analysis of the advantages and disadvantages of the ML algorithms was performed. The considered ML models included Decision Tree (DT), NB, k-means (kM), SVM, NN and deep learning (DL). Scientists have considered various possibilities for obtaining high-quality data when creating predictive models. Discussing the advantages of using the DT algorithm, it can be noted that it is fast and accessible for the user to understand the mechanism of its operation. Nevertheless, for its successful application, it is necessary to observe the rule of mutually exclusive classes and determine a clear order of attribute selection. In addition, there is a risk of retraining. Numeric attributes must comply with the law of normal distribution. The same limitations exist for the NB and k-NN algorithms, but the k-NN algorithm has a significant advantage; it is resistant to possible misplaced values in the dataset and to noise. Scientists note that the k-NN algorithm can be applied to nonlinear classification, and its model is robust. Discussing the SVM and NN models, it is impossible not to note the long time spent on training and operating the models, the risk of retraining, and the difficulties often encountered in interpreting the results. The search for the optimal network architecture often takes place through a series of tests. Nevertheless, these models are resistant to noise and missed dates, limit the risk of error, and can be used to solve both regression and classification problems. These and many other advantages are the reason for their popularity in solving problems of recognizing malignant tumors and predicting the probability of patient survival.

The data regimentation and registration on new cancer cases require a lot of human resources. To solve this problem and to provide the automated input of data on the new cases of pathologies of prostate adenocarcinoma, the authors of [8] proposed a pre-trained ML model based on the SVM method. The data collection for the study took place at the Bas-Rhin registry in two stages. First, 982 cases of pathologies in 2014 were analyzed to build predictions for 785 cases filed in 2015. Then, 2089 cases in 2014–2015 were analyzed, and 926 possible pathologies were presented in 2016. The overall accuracy for the interventions was about 97%, the SVM method was well-demonstrated on the unstructured reports of pathologies, and the main errors were associated with the patients’ place of residence. To achieve the results obtained and reduce the number of pathology reports, the authors performed a preliminary analysis of all incoming data. This approach consisted of searching the keyword “simple” in the patient’s history, as well as the exclusion of uninformative records and the removal of unnecessary characters. The R software (version 3.5) and rWeka package https://cran.r-project.org/web/packages/RWeka/index.html (accessed on 12 December 2023) were used as the tools for the implementation of the developed algorithms.

Every year, more than 600,000 cases of colon cancer are detected in the world. One of the main aims of applying ML in medicine is to build and improve the accuracy of predictive models for the development of the disease, the effectiveness of the prescribed treatment, or the occurrence of postoperative complications and re-hospitalization. In [9], the authors applied DL methods to diagnose six types of colonic mucosal lesions using convolutional neural networks (CNNs). Scientists used pre-trained deep CNN of the ResNet and EfficientNet architectures to develop an algorithm for automatic segmentation of WSI colon biopsies. The authors compared several methods and approaches for solving the classification problem and concluded that the multiple-label approach was preferable because some WSI fixes may belong to more than one class or none of them. As a result, they achieved an area under the receiver operator characteristic (ROC) (AUC) curve of up to 0.96.

In [10], E.L. Barber et al. proposed improving the accuracy of such methods based on the use of natural language processing (NLP) on unstructured data. They collected a set of medical reports with preoperative CT in women with ovarian cancer to isolate additional sign vectors. To form the required datasets, the Northwestern Medicine Enterprise (NM-EDW) repositories were used, in which patients were over the age of 18 years old and underwent surgery between 2011 and 2017. The authors reported that all language deviations had been removed to extract the target functions during data preprocessing. Unigrams and bigrams were chosen as the form of extraction since they showed the best results. Further, the authors formed three sets of NLP functions, where in the first case, a token dictionary was used together with LR using the LASSO method to reduce the dimension. The second set used all non-zero coefficients from the first case, applying principal component analysis. For the final set of functions, word2vec was used to generate patient vectors for the dictionary matrix tokens. The authors implemented two models of outcomes according to the degree of postoperative complications. For verification, 5-fold cross-validation was implemented. This reportedly resulted in a slight increase in AUC of 0.01 on average for both cases. The transfer of the obtained results into the final prediction model made it possible to achieve a 20–25% increase in the accuracy of predictions.

It is possible to predict a pathological response in gene overexpression from cancer patients receiving neoadjuvant chemotherapy (NAC). Research was carried out on breast cancer, evaluating intratumor HER2 gene expression [11]. The final model had high sensitivity of 99.3%, specificity of 81.3%, positive predictive value (PPV) of 97.9%, negative predictive value (NPV) of 92.9%, and diagnostic accuracy of 96.4%. The results show that this model may be helpful in clinical decision making. In terms of successful validation in larger multi-institutional studies, these results could be used to better select those patients who could benefit from anti-HER2 treatment.

In [12], Hsiao-Yu Yang et al. built ML prediction models for breast cancer using the breath biopsy principle. This principle helps to make decisions both during treatment and during surgery. The implemented breath biopsy system uses a sensor array to determine the structure of biomarkers and supervised machine learning methods. A carbon dioxide (CO_2_) monitoring system receives the purest alveolar air, excluding eating, oral, and smoking contaminations. If the exhaled breath from the lungs has a high enough dose of specific volatile metabolites (or volatile organic compounds, VOCs) to be detected by the sensor array, then those features can serve as biomarkers. In this study, the collected air was analyzed by the Cyranose electronic nose (E-nose) 320, a chemical vapor sensing instrument consisting of 32 chemical sensors based on nanotubes in the standard array which converts some VOCs in the breath to a sensor resistance response. Technically, gas chromatography in combination with mass spectrometric analysis is performed by the E-nose, and the analysis of respiration biomarkers with specific VOCs is collected into datasets. For example, the breast cancer pattern contains pentane VOCs as a disease marker and can be detected by the E-nose with reasonable accuracy of 90% and higher. The datasets were collected for two years (2016–2018), and a total of 899 subjects were screened and estimated. The ML model and parameters consist of the following algorithms: k-nearest neighbors (k-NN), NB, DT, NN, SVM with linear, radial, and polynomial kernels, and RF. The authors reported that the RF model surpassed the others, having the following parameters. The prediction accuracy of breast cancer in the dataset was 91%, and the AUC was 0.99 on a 95% confidence interval (CI) of ROC. PPV and NPV were the same, at 97%, respectively.

It is a well-known fact that early diagnosis of lung cancer increases the patient’s chances for recovery, as well as social and labor adaptation. The analysis of metabolomic biomarkers and changes in a metabolite pattern at an early stage provides an informative approach to the assessment of tumor progression and plays a crucial role in distinguishing the tumor stages. Such an analysis has a different degree of sensitivity, which indicates different stages of lung cancer. As a rule, if metabolomic biomarkers are detected as part of the early diagnosis, additional examination is prescribed. The process of discovering and identifying metabolites as promising biomarkers is implemented in the study of Ying Xie [13]. According to the study, six ML techniques were applied to 110 lung tumor patients at stage I and 43 healthy individuals; the best AUC value achieved was 0.923 (95% CI: 0.871–0.975), with a sensitivity of 79.6% and specificity of 93.0%. The data preparation in this study began with the metabolomic chromatographic separation of the collected assays, then liquid chromatography–mass spectrometry (LC-MS) analysis was performed using a Waters ACQUITY UPLC coupled with a 4000 Q-TRAP mass spectrometer. All LC-MS data were preprocessed and normalized (AB Analyst Software package version 1.6.2), multivariate statistical analysis (SIMCA-p14 package (Umetrics AB, Umeå, Sweden)) was applied, and principal component analysis (PCA) was selected for dimensionality reduction problem solving. Based on 46 influential metabolomic biomarkers with statistical significance for stage I lung tumor patients, the top 10 metabolic biomarkers with higher diagnostic value (AUC > 0.800) were chosen. These 10 biomarkers made it possible to clearly separate patients with stage I lung cancer from healthy persons. These data composed a training set for the k-NN, NB, AdaBoost, SVM, RF, and NN ML techniques. The authors reported that the NB, RF, SVM, and NN models showed the best prediction quality (AUC was 1.000), and the sensitivity of these models was from 0.909 to 1.000, which is enough to recognize the early lung tumor stage and appropriate for diagnosis.

Prostate cancer can be identified by investigating a specific gene, prostate cancer gene 3 (PCA3). PCA3 observation shows the overexpression of this gene in tumor tissues in 95% of cases, and thus it is an essential biomarker candidate. The authors of [14] used their original low-cost biosensors for PCA3 detection to avoid complex, specialized approaches that require polymerase chain reaction, optical changes measurement in the emission of light and fluorescence emission methods, and transcription-mediated amplification. The biosensors were fabricated using multi-walled carbon nanotubes and layer-by-layer (LbL) films of chitosan with interdigitated gold electrodes. Then, the PCA3 ssDNA probe equipped with incorporated gold nanoparticles was investigated by electrochemical impedance spectroscopy (using a PGSTAT 204, Autolab system) and UV–vis spectroscopy (using a Hitachi U-2001 spectrophotometer). The spectroscopy measurements were mapped into 2D spectrum plots by applying the interactive document mapping technique while preserving the similarity of the spectral objects in projected space. Next, a digital scanning microscope was used and a dataset for the ML training was acquired. As the training dataset contained a relatively small number of samples, the data augmentation procedure was implemented, and an eight-class classification task was formulated. Before the classification, texture feature extraction was implemented with the following techniques: Gray Level Difference Matrix (GLDM), Fourier descriptors, Complex Network Texture Descriptor (CNTD), Fractal descriptors, Adaptative Hybrid Pattern (AHP), Local Binary Patterns (LBP), Complex Network, Randomized Neural Network (CNRNN), and Local Complex Features and Neural Network (LCFNN). Three types of classifiers were used: SVM with linear kernel; Linear Discriminant Analysis (LDA); and 1-Nearest Neighborhood (1-NN). The authors reported that the maximum accuracy was 99.9 (0.3) using the LCFNN descriptor with SVM and LDA classifiers in the binary classification.

Genomic lesions involve in a variety of morphologic abnormalities, and morphologic evaluation of blood and marrow cells serves as an essential and general approach to studying myelodysplastic syndrome pathogenesis. The diagnosis of myelodysplasia (MDS) is rather difficult due to the large variety of chromosome lesions, diversity of epigenetic alterations, and complexity of processes when the blood-forming cells in the bone marrow experience sufficient transformations and become abnormal. Improving diagnosis and prognosis of MDS can be achieved with modern AI/ML models and statistical and bioinformatics tools applied to large genomic and morphologic data, according to Y. Nagata et al. in [15]. They analyzed bone marrow morphologic alterations in 1079 MDS patients and identified a total of 1929 somatic mutations with whole-exome sequencing (Illumina HiSeq 2000 device). Sequencing errors were reduced and targeted sequencing was performed. To improve data quality, the authors used ANNOVAR sample annotation and compared them with sequenced controls and mutation databases (including dbSNP138, 1000 Genomes, ESP 6500, and Exome Aggregation Consortium (ExAC) databases), in addition to filtering. Using the Consensus Cluster Plus and radomForestSRC packages in R, the authors determined that the most frequently mutated genes were TET2 (20%), ASXL1 (17%), SF3B1 (13%), SRSF2 (11%), DNMT3A (11%), and RUNX1 (10%). Based on this result and the Bayesian ML procedure, there were eight genetic signatures identified: a DT of eight subtypes for the LR MDS patients (LR-SA through LR-SH). The total accuracy of this approach for the detection of somatic mutations is up to 98.7%.

With the advent of the Clustered Regularly Interspaced Short Palindromic Repeats (CRISPR)-associated protein 9 (Cas9), it has become possible to use computer technologies to edit genes for further use in hematology and oncology. In [16], Han Zhang and Nami McCarthy describe the prospects of using the CRISPR system for genome editing to treat hematological diseases such as lymphoma, myeloma, leukemia, etc. The conducted studies have shown that the use of the CRISPR system makes it possible to identify the potential therapeutic effects of using various drugs in the treatment of lymphomas. Thus, in [17], a target gene for the therapeutic activity of NUTLIN3 was discovered. The use of CRISPR in the suppression of the oncogenic transmembrane protein MUC1-C made it easier to create knockout agents for the treatment of multiple myeloma (MM). However, there is a lot of clinical practice ahead to prove the effectiveness of this method. Therefore, aa team of scientists at Sichuan University in Chengdu working under the guidance of the oncologist Lu You [18] received permission from the ethics commission, and in 2016, for the first time, injected human cells whose genes were modified using the CRISPR/Cas9 technology. The rapidly increasing popularity of this method for the treatment of cancer provides a great number of ways to integrate CRISPR and ML to identify the CRISPR arrays using ML algorithms. In [19], the authors proposed an approach for separating the true CRISPR data arrays from false ones based on the developed CRISPRidentify tool. The developed pipeline consists of two stages: the detection and generation of candidate arrays and universal evaluation of arrays to increase sensitivity and specificity. A. Mitrofanov et al. in [19] identified new training sets, based on which they then trained a classifier to evaluate candidate arrays. Six different datasets were used. A feature vector was generated for each candidate array, which was used to evaluate the pipeline. The Extra Trees classifier from the Python Scikit-learn package https://scikit-learn.org/stable/ (accessed on 12 December 2023), which was integrated into the pipeline, was used to conduct the assessment. As a result of a number of tests, the Extra Trees method surpassed the classifiers based on NB, SVM, k-NN, etc., with a median accuracy of 0.91. In the conducted test, CRISPRidentify was able to detect 147 previously annotated arrays. The developed tool is able to detect the arrays with a limited number of repeating separators, as well as to create dynamic filtering criteria based on the data received during training.

The molecular genetic somatic mutation research in cancer plays an essential role and the presence of mutated genes that belong to the Ras subfamily (e.g., KRAS, NRAS, HRAS) allows for the detection of several cancer types. The Ras subfamily genes can normally exist in the inactive form and encode a cytoplasmic protein involved in intracellular signaling from growth factor receptors. For example, in the vast majority of cases, constant activation of KRAS and NRAS genes leads to malignant mutation of the cells and activates the EGFR-RAS-RAF pathway. This pathway signals tumors in such primary cases as pancreatic adenocarcinoma, lung adenocarcinoma, thyroid carcinoma, and skin cutaneous melanoma. In [20], Way et al. proposed a method of classification and prognosis of Ras activation in cancer using ML and the Pan-Cancer Atlas data. At that time, the data in The Cancer Genome Atlas (TCGA) Pan-Cancer Atlas included 9075 tumors across 33 different cancer types, but now the number of tumor cases included in the Atlas is over 11,000. The data for the ML model training were collected from three sources: the GISTIC2.0 source, cancer cell lines, and Illumina RNA sequencing. The ML model was trained on those data after preprocessing and filtering. The model is elastic net LR with penalties. The elastic net is a regularized regression algorithm that combines both the LASSO and Ridge techniques to improve regularization through removing weak variables altogether, as with LASSO, or reducing them closer to zero as with Ridge. The authors reported high performance of their ML model, with 84% AUC, in predicting Ras pathway activation metrics.

Recent years have witnessed the success of CNNs, driven by the development of DL, dramatically increased computing power of GPUs, and the emergence of large volumes of labeled datasets in visual recognition problems. As a result, DL and CNN have become increasingly popular in CV in various fields [21], including diagnostics and medicine. The most common examples include classification of skin lesions [22], histological classification of breast cancer [23], and mortality assessment by chest radiographs (lung cancer). ML is increasingly used to solve the problems of segmentation and classification of cancer cells in various locations in the body; new systems are being developed and improved that can detect the presence of pathological cells and classify them using a loaded image of scanned glass with histology [24,25,26,27,28].

One of the examples of using the ML models to solve the problem of skin cancer recognition was the development of the “SkinVision” medical service https://www.skinvision.com (accessed on 5 February 2024). The developed application combines AI technologies and the experience of highly qualified oncologists in order to provide recommendations for the prevention of skin cancer and timely detection of diseases. Among the assessed signs are the shape, texture, and color of skin formations. It has been proven that the use of the algorithm in 95% of cases makes it possible to detect the disease at an early stage, but the control of a human specialist is still a prerequisite.

An important application of the ML methods is the study of attenuation correction and image restoration in PET. In [29], Tonghe Wang et al. confirm that in PET, an accurate quantitative assessment of the absorption of indicators is of great clinical interest. Incorrect solution of the problems of inaccurate attenuation and scattering of photons can lead to a deterioration in the accuracy of the PET diagnostics. The authors claim that in recent years, sharing Magnetic Resonance (MR) with PET has become a promising alternative to PET/CT due to the excellent anatomical visualization of soft tissues without ionizing radiation. The automatic classification and manually drawn contour methods are limited by the inaccurate prediction of air and bone. These methods have been replaced by other techniques, such as warped atlases of the MR images labeled with known attenuation factors to the patient-specific MR images. Their effectiveness strongly depends on the registration algorithms. The authors emphasized that image post-preprocessing and the use of noise regularization during reconstruction allowed them to compensate for the differences between neighboring pixels to reduce the noise level in the image. The study suggests two solutions, depending on whether only a PET scanner was used or PET/MR images were obtained. In the second case, the authors analyze the advantages of using the convolutional auto-encoder (CAE) to generate CT tissue labels from the source images. In addition, an RF method is considered for training a set of decision trees. These methods have proven their effectiveness. The MR images are not available in the PET scanner. The most common networks used for processing PET data are GAN and U-Net. The authors consider using the cycle-constant GAN (CycleGAN) method, which showed excellent performance in most of the evaluated indicators and less bias in defeats. In addition, high-resolution PET can be used to visualize and accurately measure the concentration of a radioactive indicator in small structures.

Endometrial cancer (EC) is the most common gynecologic cancer in industrialized countries. About 15% of patients develop recurrence with limited treatment options and poor survival. To improve patient survival and to choose between the most invasive surgical procedures or drug therapy, the reliable identification of patients from different risk groups is required. To minimize the risks of cancer recurrence, new methods of image analysis are needed.

In [30], E. Hodneland reviewed a fully automated approach to primary tumor segmentation in endometrial cancer using three-dimensional CNNs. With this approach, the estimates of tumor volume and the accuracy of CNN-based segmentation are comparable to the segmentation results obtained by radiologists. Automated and accurate CNN tumor segmentation provides new opportunities for full-scale operational radioma tumor profiling, allowing for prognostic markers, which makes oncology treatment more personalized for each patient.

Preoperative MRI in patients with endometrial cancer provides information about the spread of the tumor, which is necessary for the choice of a surgical procedure and drug therapy. In addition, the full range of tumor MRI can provide the radioma signatures of the tumor, which are important for the selection of the optimal treatment regimen for a particular patient. The sample for the CNN training consisted of 139 patients, while expert physicians significantly disagreed on the placement of the primary tumor in 14% of cases. CNN allows for the automatic extraction of tumor volume and tumor texture characteristics, which makes it possible to obtain better prognosis and individualization of therapeutic strategies for endometrial cancer. The results obtained were comparable to the human expert level, which was confirmed during the Wilcoxon signed rank test. Manual segmentation of the tumor to calculate the volume and study the textural characteristics of the tumor is a time-consuming process. E. Hodneland et al. demonstrate the use of two-dimensional U-Net convolutional networks to solve the segmentation tumor issues. There are many powerful frameworks and platforms that allow specialists to implement DL in 3D medical imaging. The automatic segmentation method has the potential to detect systematic changes in tumor volume in response to various treatment strategies. Such potential has high value in clinical drug trials. There were no significant differences in the conducted Intraclass Correlation (ICC) estimates. This suggests that the network performs functions at a level comparable to radiologists.

CV development in medicine has recently opened up many new avenues for solving problems in the field of cardiovascular diseases, where echocardiography is uniquely suited to DL as a simple and effective method for collecting and interpreting data [31]. DL helps pathologists to conduct histological analysis of tissue samples, which is subjective in nature, and different perceptions of pathologists can lead to misdiagnoses [32]. The latest advances in technology have made it possible to create new scanners that capture histological specimens with high resolution and obtain WSIs [33]. This, combined with advances in AI, has led to the research and commercialization of AI-based digital histopathology [34]. James T. Grist et al. describe the methods of RF, l-NN, and AdaBoost, which were used to create the model [35]. The univariate classifier obtained 85% accuracy with AdaBoost. Using the SVM method had the highest precision in discriminating between high- and low-grade tumors. The study showed that the combination of the multiparameter MRI, the ML methods, as well as single-factor analysis can be used to distinguish both highly differentiated and low-differentiated brain tumors in children, as well as to classify tumor types with a high accuracy of about 85%.

The study [33] by Evans et al. provides examples of medical data visualization in cardiology, dermatology, and pathology. The authors note that CV enables solving medical problems such as screening, segmentation of pathologies, predicting outcomes, as well as disease monitoring. The most important DL technique used to solve these problems is CNN. It hardcodes the key feature of images—invariance. The study of medical data is associated with specific problems. Various methods are used to solve them. The authors describe the use of supervised learning, which uses data points and data labels.

Glioma is one of the most common primary brain tumors. ML is used to visualize gliomas and can act as a non-invasive assessment of the molecular characteristics of gliomas. After such an assessment, the data are sent to the laboratory for further clinical trials. It is also noted that the segmentation of tumor volumes in PET can be automated by implementing CNN. Specialists face difficulties in diagnosing children’s brain cancer, because the MR scanning method is ineffective in the brain areas with poor magnetic field homogeneity and small lesions. T.C. Booth et al. [36] discuss the methods of diffusion-weighted imaging (DWI) and dynamic susceptibility contrast imaging (DSC), which are used in children brain tumor diagnostics. DWI creates images weighted by the water velocity in a given voxel. DSC displays the dynamics of contrast media using rapid imaging techniques (EPI and PRESTO). Supervised machine learning utilizes mean apparent diffusion coefficient (ADC) or mean cerebral blood volume (CBV) data features and high and low grade of tumor types as classes for mathematical algorithm training and assigning datasets to classes. The applications of supervised learning to oncological medical imaging have commonly utilized single measures of the tumor microenvironment between tumor types. F-statistics were used to evaluate the discriminant ability of classifiers. The authors in [36] conducted controlled ML using the Orange toolbox in Python (version 3.6) and a single NN.

The Wisconsin diagnostic breast cancer dataset (WDBC) has remained in demand for many years. It consists of a total of 569 records classified into two categories: malignant (212) and benign (357) tumor cell cases. A detailed description of WDCB that contains 10 essential real-valued features (radius, texture, perimeter, etc.) computed for each cell nucleus makes this dataset an excellent candidate for ML model training and estimation. P. Gupta and S. Garg, in their study [37], applied six different models to WDBC, including k-NN, LR, DT, RF, SVM, and deep learning ANN (DL_ANN). They estimated statistical metrics for the above-mentioned models and reported on the following results. All six ML models trained on WDBC had high accuracy of over 95%, but in this case, the preferred ML model was the DL_ANN model, having an accuracy of 98.24%, and the precision, recall, and F1-score for this model were over 98%, too.

R.T. Lewinson et al. developed an ML neural network prediction model for the prediction of cutaneous immune-related adverse events (irAEs) from anti-PD-1 therapy [38]. This model was trained on the dataset of 143 metastatic melanoma and 339 non-small cell lung cancer patients treated with anti-PD-1 therapy. The authors used a three-layer fully connected feed-forward neural network, then trained it with 85% of the collected dataset and tested it with the remaining 15% of the dataset. The model showed promising predictive results, including an AUC of 76.5% and overall accuracy of 78.1%. It was concluded that ML models could be applied to the identification of at-risk patients for early dermatologic interventions.

A.M. Bur et al. [39] proposed models for predicting pathological nodal metastasis in oral cavity squamous cell carcinoma. The datasets for the ML models were extracted from the National Cancer Database, totaling 1961 cases with confirmed tumors and 71 patients treated at a single institute. The former dataset was split into two parts of 1570 training cases and 391 testing cases, respectively. The latter dataset was split into a {51:49} percentage proportion. The authors developed several ML models for processing those datasets to predict pathologic lymph node metastasis. The following ML models were implemented: LR, DF, kernel SVM, and Gradient Boosting Machine (GBM) algorithms. The best result for the NCDB dataset was the DF classification model, with an AUC of 0.712 (sensitivity of 0.753 and specificity of 0.492), and the best result for the SI dataset was the DF classification model, with an AUC of 0.840 (sensitivity of 0.917 and specificity of 0.576).

R. L. Gullo et al. [40] provided an overview of the early prediction of response to NAC with an ML model. One can consider NAC as a potential standard that is widely used to treat several subtypes of breast cancer. Generally, early identification and prediction of breast cancer demonstrate decreasing patient mortality, and MRI is the most applied technique for this purpose. Owing to a great number of such features as texture, shape, color gradings, and others, the MRI images could be one of the essential candidates for the so-called radiomics signature. Along with proper labeling of MRI images in the radiomics signature, ML approaches could be applied to the classification and prediction of tumor changes and patients’ health parameters. The authors considered specific ML classifiers for the prediction of residual cancer burden, recurrence-free survival, and disease-specific survival, namely linear SVM, LDA, LR, RF, SGD, DT, AdaBoost, and extreme gradient boosting (XGBoost). The variables between integrated clinical and MRI data frequently have latent and non-linear relationships. Thus, the ML algorithms are supplemented by statistical methods, for example, Bayesian LR. When clinical and MRI features are combined, they demonstrated a better accuracy of 0.86 (95% CI 0.71–0.96) with a sensitivity of 0.88 (95% CI 0.71–1) and a specificity of 0.82 (95% CI 0.56–1). Predictive response to NAC is usually improved when one of the DL techniques is applied. The most used DL architecture is CNN consisting of convolutional, fully connected, dropout, max pooling, and classifier layers. The accuracy of the three-class prediction of NAC can reach up to 87.7% ± 0.6% with a specificity of 95.1% ± 3.1%, and a sensitivity of 73.9% ± 4.5%.

The balance between prognostic biomarkers of adenocarcinoma and squamous cell carcinoma (SqCC) is not in favor of the latter, since SqCC biomarkers have been studied less intensively. In the study by Y. Koike et al. [41], a total of 135 peripheral SqCC confirmed cases were analyzed to develop ML models for classifying and determining tumor, stroma, and necrosis areas. They used an ML approach to the prognosis of the stroma component significance in the histomorphometric analysis of patients suffering from lung SqCC. The dataset was prepared by converting ematoxylineosin tissue glass slides into high-resolution digital data using the Hamamatsu Photonics NanoZoomer 2.0 system, and then the images were processed with color-based segmentation through kM clustering implemented by MATLAB. As a result, the described approach provided the exact calculation area of the cancer cell component, the cancer cell necrosis component, and the stromal cell component. The authors reported that the ML approach confirmed the prognostic significance of the ratio of stromal cells in the prognosis of lung SqCC.

PET coupled with CT can serve as an integrated tool providing holistic quantification of tumor metabolic activity with tissue density and structural property description. The resulting image radiomics biomarkers and volumes of interest of images are essential quantitative predictors for high-risk human papillomavirus (HPV) infection and were studied by S.P. Haider in [42]. A complete set of PET/CT features accounting for 1317 PET and 1317 CT features per lesion of a total of 190 patients with HPV-associated oropharyngeal squamous cell carcinomas was screened. The authors developed the locoregional progression model as the ML method and utilized the combined set of radiomics and PET/CT features as a predictor for prognosis. The study reported that the developed ML predictor had median (interquartile range) C-index of 0.76 (0.66–0.81; *p* = 0.01).

## 3. Treatment Planning

Cancer treatment planning is a difficult issue. The treatment for various types of cancer is not only specific, but often has a number of side effects. In addition, assistance in the treatment of cancer requires a multidisciplinary approach and the involvement of various specialists: surgeons and radiologists, persons caring for patients during their rehabilitation, etc. Moreover, the emergence of new scientific discoveries in the field of oncology research leads to a change in existing paradigms of disease treatment. Finally, the types of cancer treatment assistance are divided into initial, curative, supportive care, and help with cancer relapses, as shown in Figure 2. All of the above points create significant difficulties in using AI for cancer treatment.

On the other hand, the application of ML in healthcare is proving its worth and validity every day. Constantly growing volumes of data, reducing resource costs, improving the quality of prescribed treatment and patient care, timely diagnosis of diseases, and other advantages allow one to gain a positive effect from the introduction of ML and Big Data analysis as essential technologies for helping medical specialists. However, such integrations require highly skilled computer scientists, making it difficult to deploy such support systems everywhere. Moreover, healthcare professionals often lack ML skills. J. Waring et al., in their review paper [43], consider automated ML (AutoML) and a comparative analysis of existing solutions and platforms. The overview is based on 101 sources in AutoML and highlights the major limitation of this technology; it is not efficient enough to work with large-scale data. Both individual elements of AutoML optimization and solutions that allow tracing the entire cycle are considered. The pivot tables of automated function design tools, the AutoML Pipeline optimizers, are presented, and the performance of Neural Architecture Search (NAS) algorithms based on CIFAR- 10 is evaluated.

The use of AI in precision oncology allows for the analysis of tumor genomic data, providing information about potentially effective options in next-generation tumor sequencing assays. It has become possible to predict the type of tumor based on the targeted panel of DNA sequence data. This, in turn, allows specialists to optimize the patient’s treatment strategy.

Genomic screening makes it possible to more correctly organize the treatment of patients based on their genetic factors. The investigation of individual genetic factors is enabled by the advent of next-generation sequencing (NGS) technology. A detailed review of NGS and the possibility of its integration with Big Data technologies, ML, and visualization of medical diagnostic data has been conducted in [44]. Timely qualitative analysis of large data sequencing provides early diagnosis of diseases. This is enabled by the accurate forecast model, the Rowan biomarkers, and the identifiers for drug discovery, which specify the strategy for effective cancer treatment for each patient individually. The authors present an overview of the existing NGS platforms with the estimation of the output data amount and the cost of the study, which makes it possible to reflect the trend in the development of precision medicine methods in the field of DNA and RNA sequencing. One of the important components of their research is the expression of medical imaging as a useful tool for detecting, monitoring, and predicting the development of cancer. They conclude that the integration of existing strategies with AI algorithms could significantly expand the capabilities of healthcare.

The paper in [45] discusses the possibility of using ML and DL in matters modulating pharmacological agents to treat cancerous diseases. This article reveals the current potential in solving the problems of developing new and using existing drugs. The importance of in silico structural methods for expanding the possibilities for the synthesis of new drugs is emphasized, among which molecular docking stands out, predicting and assessing priorities in the interaction of a target and a drug. This method is integrated with ML and DL technologies and can significantly improve the accuracy of molecular docking. In addition, the treated and other structural virtual screening techniques have been processed, such as cheminformatics modeling-based ligands using machine learning and proteochemometrics simulation with the use of ML. N.T. Issa et al. analyzed ML capabilities to predict drug–phenotype associations and indicated the applications of large-scale transcriptomics datasets. Based on current advances in natural language processing, Electronic Health Record (EHR) datasets have been evaluated to predict the potential side effects of medication. The paper provides recommendations on the use of ML and AI methods for repurposing drugs. In conclusion, the authors noted the possibilities for the integrated use of AI to solve fundamental problems in creating new and using existing drugs for new types of cancer.

Molecular profiling of tumors is an intensively developing approach that allows targetable identification of malignant somatic alterations and gives rise to a set of diverse strategies in precision oncology. Precision oncology uses genome-scale omics data and next-generation sequencing molecular analysis in addition to the data from large-scale pharmacogenetic screening cell lines and patient-derived xenografts to discover molecular targeting drugs affecting targeted mutated genes. Nevertheless, the problem of targeted drug development remains complex and non-trivial. R. Miao et al. [46] proposed a tumor drug and drug sensitivity prediction model based on the precision oncology approach combined with statistical and ML models. Their approach includes the following steps. The first step was collection from gene expression libraries (Genomics of Drug Sensitivity in Cancer Project, Affymetrix CEL files), the Cancer Cell Line Encyclopedia, and the data from the cancer drug-sensitive genomics pharmacogenomics studies. Next, the data were preprocessed to extract genomic state markers as targets for molecule-specific drugs. A total of 140 drug sensitivity experiments were analyzed in 624 cell lines with 71 gene mutation status and 22,215 probe expression values. The second step was preprocessing the dataset by Robust Multi-Array Averaging, converting the data with two logarithmic operations and single peak testing of the selected gene points on the data. This operation showed that the collected data had a bimodal distribution and were statistically significant in representing a large number of gene variations. After this test, the initial 22,215 probes were reduced to 1595 probes with significant multimodal characteristics and selected as the features for predicting the drug sensitivity. Screened potential genes allowed for choosing targeted drugs, and only 10 of the 53 candidates were approved for clinical application in genomic test instructions and mutation detection of BRCA1/2, EGFR, ERBB2, ALK, and BCR-ABL. The authors experimented with five ML models (RF, Elastic Net, Linear-SVM, PLOY-SVM, and RBF-SVM) and reported that the best result was the RF ML model, which had an average sensitivity of 0.98 and an average specificity of 0.97, and an average accuracy of 0.978. The others had no values mentioned exceeding 0.74.

ML techniques can be successfully applied for prostate low-dose-rate brachytherapy planning instead of the manual conventional planning procedures. In the study by A. Nicolae et al. [47], the authors collected a large dataset with high-quality postoperative prostate images at day 30. After feature extraction from the DICOM file format, the new contours allowed for the application of reinforcement learning, and the initial model was adjusted to improve its performance and accuracy. The developed prostate implant planning algorithm was tested on 41 consecutive patients who underwent I-125 LDR monotherapy and showed high sensitivity to excluding overdose or underdose therapy.

Radiation oncology supplies techniques that must be carefully planned to avoid side effects for patients. Stereotactic body radiation therapy (SBRT) is among them. The treatment planning should be fractioned, dosed, and assessed through a patient’s full dose–volume histogram (DVH) profile. X. Pan et al. [48] applied advanced ML techniques for planning prostate SBRT with reduced treatment-related toxicity. They involved 86 patients who underwent prostate SBRT, utilized ensemble ML algorithms, and verified selected models on 26 external cases as the test models. The authors used boosting (Gradient Boosted DT and Adaptive AdaBoost) and RF ensemble learning technologies. The investigation of the three abovementioned ML models showed that the Gradient Boosted DT outperformed the others for each of the urinary and rectal domains at 3 and 12 months after SBRT. The authors reported that the best AUC results were for the gradient boosting decision tree (GDBT), with a cumulative DVH (urinary irritation) of 0.79, cumulative DVH (urinary incontinence) of 0.87, and rectal toxicity of 0.64.

Z. Yang et al. [49] developed a DL framework and complementary statistical methods to predict the relationship between prostate radiation therapy and patients’ quality of life. The life quality data were collected after prostate cancer radiation therapy with a 14-question survey with added computed tomography images in de-identified form for a group of 52 patients. Then the dataset was augmented with the curvature-based image registration technique implemented in MATLAB and amounted to 1326 images. The image regions that had high radiation dosage were marked and outlined. The authors developed a deep NN with three-convolution-layer architecture, exponential linear units, and SoftMax classification layer to classify images of patients with or without symptoms of high radiation dosage. Due to relatively small training (39 cases) and testing (13 cases) datasets, overfitting was noticed. Thus, transfer learning was implemented to improve classification performance with autoencoder implementation to pre-train the CNN on the augmented unlabeled data. The CNN results were compared with statistical methods, including *t*-test, ANOVA, and LR. They reported that the best accuracy of the developed prediction model was 84% for the rectum model and in the range of 23–53% with a median accuracy of 38%.

An ML-based approach for predictive values of gene expression profiles in formalin-fixed paraffin-embedded tumor biopsies was proposed by M. Wiesweg et al. in [50]. The authors performed digital expression gene analysis on 30 potential reference genes using the NanoString nCounter platform and the PanCancer Immune Profiling panel consisting of 770 genes involved in an immune response. The authors performed digital expression gene analysis on potential reference genes (CYLD, CXCL11, MYD88, PRPF38A, HLA-E, STAT6, CD59, and STAT2) using the NanoString nCounter platform and the PanCancer Immune Profiling panel consisting of 770 genes involved in an immune response. A total of 127 patients were screened, and then 55 generated admissible nCounter reads were collected in the training cohort. This dataset underwent RNA feature selection using an ensemble approach with random subsampling and regularized with LASSO/L1 regression. The following ML models were implemented in the Python language: SVM (scikit-learn SVC for classification and SVR for regression), RF (scikit-learn RandomForestClassifier and RandomForestRegressor), LR (scikit-learn LogisticRegression for classification and ElasticNet for regression), and GB (xgboost XGBClassifier and XGBRegressor). It was concluded that ML models could be useful for predicting the positive identification of a group of patients benefitting the immunotherapy.

## 4. Patient Survival Prognosis

Modern technological advances in computer science include artificial intelligence and promote the use of ML algorithms in medical diagnostic and research, which is a strategic direction in the development of decision support systems in oncology, as shown in Figure 3.

This integration can help detect and classify pathologies, create or modify diagnostic test descriptions, and predict disease progression and patient response to therapy. Nevertheless, it is fraught with several problems and limitations in the actual application of the technology in a clinical setting. For a qualitative assessment of possible applications in the study of R. Cuocolo et al. [51], a review of modern literature was carried out, including 90 relevant sources. The summary tables of the ML algorithms are presented. They are used to assess the risks of oncological disease detection, and create descriptions of pathologies, as well as classify and predict types and stages of diseases, indicating a specific goal and type of oncological pathology. All indicated data were accompanied by reduced accuracy, specificity, and sensitivity of the models used. For each of the cases, the limitations for real application were given, and the statistics of the application of ML methods on the training and test dataset were shown, indicating the quantitative and qualitative characteristics of the latter. The possibility of constructing predictive models of patient survival based on ML methods was assessed, as well as the use of AI to develop targeted methods of treating cancer. The data in [51] have a clear understandable structure, giving comprehensive explanations of the current state of the art on the application of ML in oncology in a clinical setting.

Radiomics can also be used to develop a robust radiomics-based classifier that is capable of accurately predicting the overall survival of renal cell carcinoma patients for the prognosis of clear cell renal cell carcinoma. This signature may aid in the identification of high-risk patients who need additional treatment and follow-up regimens. For this model, the AUC, accuracy, sensitivity, and specificity with a 95 percent confidence interval were 0.95–0.98, 0.93–0.98, 0.93–0.96, and 1.0, respectively [52].

One of the determining factors of surgical intervention and the degree of resection in meningioma is the preoperative determination of its consistency. Thus, the study of S. Cepeda [53] aims to develop a predictive model for determining the parameters of meningioma consistency based on preoperative medical diagnostic studies, including MRI and ultrasound elastography (IOUS-E) using ML classifiers. The data from 18 patients operated on in the period 2018–2020 were selected as the initial dataset. All cases were followed by IOUS-E and T 1 WC MRI diagnostics. Preliminary processing of the diagnostic data consisted of the decomposition of the DICOM electrogram format into hue–saturation–brightness (HSB) with subsequent manual tumor segmentation, as well as the removal of MRI radioma signs using the open-source LifEx version 6.0 software and the subsequent automatic segmentation using ITK-SNAP. The softness meningiomas performed value average tissue elasticity (MTE) was chosen as the reference value. As a predictive model, a DT was used, which determined the threshold value based on the generated groups of intraoperative MTE characteristics and made it possible to determine the consistency of the meningioma, where Range version 3.26 was used to optimize the algorithm. The classification system was based on six different ML algorithms, including LR, NB, kM, RF, SVM, and MLP. The models were evaluated using the AUC and reached values from 0.699 to 0.974, with an accuracy of 61–94% and a precision of 60–95% on each algorithm used. The best results were demonstrated by the combination of the Information Gain and ReliefF filters with the NB model, in which the AUC was 0.961, with an accuracy of 95%, corresponding to a classification accuracy of 94%.

The great advantage of using ML in processing data is its ability to evaluate the impact of possible surgical intervention and potential risks of developing distant metastasis. It can provide biological information that cannot be determined from conventional MRI characteristics. Using a retrospective dataset, researchers [54] investigated an ML model based on radiomics, which is favorable for predicting the likelihood of distant metastasis from soft-tissue sarcoma. It was shown that the AUC in the validation set of patients with soft-tumor sarcoma with distant metastasis may be 0.9510. That means that the probability that the mechanism will rank an instance as a true positive is higher than the false positive rate. This may help with clinical decision making to lower the burden of costs from follow-up examinations and anxiety related to false positive results by understanding the most common organs of distant metastasis.

Radiomics has emerged as a promising clinical tool for a variety of clinical problems, including drug development, clinical diagnosis, treatment selection and implementation, and prognosis. Nowadays, radiomics is primarily used to calculate progression-free survival or overall survival, predict tumor sensitivity to treatment, and perform classification tasks that do not fall into one of the above categories. These techniques have the potential to improve patient care in future AI-based radiotherapy workflows by developing personalized medicine and prescribing personalized doses [55].

Radiation therapy coupled with chemotherapy is frequently complemented by radical surgery in gastric cancer treatment. Nevertheless, the question of the toxicity and dosage of radiation therapy constantly arises. In this regard, there is an increasing interest in the use of AI technologies and ML for the prediction and evaluation of the overall survival of patients who undergo radical surgery, radiation therapy, and chemotherapy. M. Akcay et al. [56] assessed the applicability of several ML algorithms to predict the overall survivability of patients with confirmed gastric cancer, peritoneal recurrence patterns, and distant metastasis after treatment. The authors involved several ML algorithms, including LR, MLP, XGBoost, SVM, RF, and Gaussian Naive Bayes (GNB). The dataset was collected from the data of 75 cases of patients with gastric cancer undergoing radiation therapy and chemotherapy at the Department of Radiation Oncology of Eskisehir Osmangazi University Faculty of Medicine (Turkey). The authors avoided an imbalance in the collected dataset and applied the synthetic minority resampling (SMOTE) technique. The ML algorithms and statistical functions were implemented using the Python and Scikit-Learn library. The authors reported that the best prediction performances of overall survivability, distant metastases, and peritoneal metastases were obtained with GNB, with an accuracy of 81% (95% CI, 0.65–0.97, AUC = 0.89), XGBoost, with an accuracy of 86% (95% CI, 0.74–0.97, AUC = 0.86), and RF, with an accuracy of 97% (95% CI, 0.92–1.00, AUC = 0.97), respectively.

ML-based MRI radiomics can be used to improve the accuracy of a prediction model. Using various datasets, the researchers tested a model that was aimed at distinguishing aggressive from non-aggressive papillary thyroid carcinoma. In the testing cohort, a prediction model using only clinical characteristics achieved an AUC of 0.56, which is low and close to a random result. The AUC of the ML-based MRI radiomics was 0.92, indicating improved performance over the clinical characteristics model. A new model that combined clinical characteristics and radiomic features was similar to the model that only used radiomic features. This finding is promising for a non-invasive method of assessing cancer aggressiveness. Assessing the aggressiveness of papillary thyroid carcinoma before surgery may assist clinicians in planning a surgical approach, such as lobectomy, subtotal thyroidectomy, or total thyroidectomy, with or without neck dissection [57].

The main strategy for using ML to solve the problems of radiation oncology in most works published on this topic is to simplify and reduce the problems to the form of a binary classification, which can cause unreliable research results. In addition, there are several problems that can be solved in terms of multiclass classification. In the study by A. Chatterjee et al. [58], the authors provided the facts that enabled them to evaluate the results for several studied classes reliably and the possibility of using surrogate markers in a clinical setting. The authors reported that, unlike binary classification, the AUC and the correlation coefficient could not serve as reliable indicators of performance for multiclass classification in a clinical setting. The validity of their arguments was confirmed using the Monte Carlo model (MC), which showed that the capacity prediction characteristics decreased with the amounts and classes of outcomes. In the second part of the study, devoted to the use of surrogate markers in real conditions, the MC analysis was performed by calculating five scale factors and adding noise to the surrogate marker for each patient from five cohorts with different prescribed treatments. The results were obtained with the statistical differences, separating groups of patients when using a surrogate marker according to the criterion rank-sum Wilcoxon (*p* < 0.001).

Feature-based and DL-based radiomics is being increasingly evaluated in neuro-oncology [1]. AI can be useful for the differentiation of treatment-related changes from tumor progression in patients with gliomas and brain metastases (AUC of 0.73–0.96), which is a critical clinical question. The addition of molecular markers to histology is recommended to define brain tumor entities but it damages the tissue, and the core biopsy specimen cannot be taken very often. A noninvasive method is a good solution when tissue samples are not accessible. It is possible to predict the isocitrate dehydrogenase (IDH) genotype (AUC of 80–96), loss of heterozygosity of the 1p/19q chromosome arms (AUC of 67–96), and the O6-methylguanine-DNA methyltransferase (MGMT) promoter methylation status (AUC of 80–95). It is molecular markers on which treatment strategies are based. Furthermore, it was demonstrated that the expression levels of the tumor cell proliferation marker (Ki-67) can be used to calculate or predict proliferative activity in gliomas (AUC of 76–92). Aside from that, several studies since 2016 have evaluated radiomics for the determination of grades from the World Health Organization (WHO) classification of central nervous system tumors (AUC of 87–97).

Methylation of the MGMT promoter could also be used to predict overall survival. It has been shown that the age and MGMT methylation index are the most important factors for predicting overall survival in the chosen models. Radiomic features provide additional information beyond the scope of human visual perception, which is considered to be valuable in the prediction of glioblastoma OS and progression-free survival, and there is relatively limited literature on this topic [59].

One significant advantage of in silico studies is that researchers have no restrictions on the parameters or features they can use to find correlations. A radiomics study identified optimal ML methods for the radiomics-based prediction of local failure and distant failure in advanced nasopharyngeal carcinoma [60]. They tested different parameters and features in combination to divide patients into three groups: those who experienced local failure, those who experienced distant failure, and those who did not experience either local or distant failure during the follow-up period. It helped to identify new parameters that are representative enough in this issue to become new imaging biomarkers to assist clinical decision making. Quantitative radiomics has the potential to significantly improve cancer care by facilitating the implementation of personalized medicine.

Neuro-oncology is gradually expanding its field of knowledge using ML models. C.A. Sarkiss and I.M. Germano presented a systematic review [61] devoted to using ML-based algorithms in neuro-oncology. They extracted some information about ML application from 29 studies and the data on a total of 5346 patients who were screened for neuro-oncology diseases, such as glioma, glioblastoma, brain metastases, etc. The studies mentioned above reported on a set of ML algorithms including SVM, RF, Gaussian NB, LDA, DT, gradient boosting (GB), and adaptive GB. Then, the ML algorithms were clustered by the task and application into three categories: outcome predictors, image analysis, and gene expression. The authors reported that the outcome prediction was used in 12 of 29 studies with a sensitivity range of 78–98% and specificity range of 76–95%. The image analysis was conducted in four studies. Finally, the remaining studies were devoted to ML-based gene expression analysis and had an accuracy ranging from 80% to 93%. The study in [62] examined automatic contour scoring using fit indices and supervised machine learning. The issue is relevant because the peer review of the target volume (TV) and organ at risk (OAR) contours during radiation therapy planning is conducted visually and can be time-consuming, and the result can vary greatly among observers. The best result was obtained for stomach images with an overall prediction accuracy of 96% (using the Vector Machine and Ensemble models); the worst result was shown by LR—76% for liver images. The considered method of automatic contour estimation makes it possible to reduce the review time, eliminate gross errors in contour determination, and can also be used for training doctors.

R.O. Alabi et al. [63] compared the nomogram technique with several ML algorithms to estimate the performance of such approaches in the prediction of tongue cancer survival outcomes. They obtained the dataset from the National Cancer Institute through the Surveillance, Epidemiology, and End Results Program of the National Institutes of Health, including 7649 cases where patients were diagnosed with positive histologically confirmed tongue cancer. Then the dataset was used for labeling and predicting nomogram building. The nomogram accuracy with surgical treatment was 66% and the nomogram accuracy with radiotherapy was 60.4%. The sensitivity and specificity were 1 and 0.62 for the first case and 1 and 0.55 accordingly for the second case. The authors trained seven ML models (LR, NB, SVM, ANN, Boosted Decision Tree, Decision Forest, Decision Jungle) in Microsoft Azure ML Studio 2019 to predict an overall survival month using 5-fold cross-validation, splitting the dataset in {80:20} percentage proportion for training and validation. The best accuracy of 83.1% was achieved with the ML model of Boosted Decision Tree, with an AUC of 0.9. The authors reported that the comparative performance metrics of the ML algorithms were higher than the nomogram approach, and the accuracy, sensitivity, specificity, and F1-score were 88.7%, 1, 0.87, and 0.66, respectively.

In [64], A.V. Karhade et al. demonstrated the applicability of ML techniques to prognosticating survivability with confirmed rare tumors. One of them is spinal chordoma, and due to the rareness and relatively slow change of this tumor type, long-term prediction is not a trivial task. The authors obtained statistics for a long period of time (over 15 years), and estimated survivability for 265 patients, showing a mean value for 5-year survival of 67.5%. Then, the conventional statistics operations were compared with four ML models, including BDT, SVM, Bayes Point Machine, and NN. These algorithms were trained, assessed by a 10-fold cross-validation technique, and a web application using Microsoft Azure (Microsoft Corporation, Redmond, WA, USA), R Studio (version 1.0.153), and Anaconda with the Python distribution (Anaconda, Inc., Austin, TX, USA) was implemented. A.V. Karhade et al. concluded that the Bayes Point Machine showed the best predictive capability, having AUC = 0.801. They also pointed out that the ML models in long-term survival predictions had high potential, but a weak explanatory ability.

It is challenging to avoid subjective risk estimation in patients’ mortality from cancers while receiving febrile neutropenia chemotherapy, but the inpatient mortality risk prediction is much harder. X. Du et al. showed in [65] that ML models could provide a non-subjective prognosis in mortality risk prediction. They used a dataset from the HCUP’s National Inpatient Sample and Nationwide Inpatient Sample on the patients with diagnosed cancer and picked the data with febrile neutropenia, totaling 126,013 adult admissions. These data were complemented with in-hospital information, and the descriptive statistics were calculated. Then, the authors developed two groups of models: linear models including Ridge LR and linear SVM, and non-linear models including GBT and six-layer NN models. After the model tuning and three-fold cross-validation process, the models demonstrated surprisingly the same AUC, sensitivity, specificity, precision, recall, and F1-score. Those parameters were 0.92, 0.81, 0.88, 0.84, 0.70, and 0.74, respectively. The GBT model was slightly different from the others, but it was no more than one hundredth better.

A prognostic system for endometrial cancer using ML was proposed by A.M. Praiss et al., utilizing the Surveillance, Epidemiology, and End Results (SEER) database and Ensemble Algorithm for Clustering Cancer Data (EACCD) for their research [66]. ECCD made it possible to determine dissimilarities in defining and computing on the SEER data and then applying the hierarchical clustering analysis while visualizing the relationship between survival and prognostic factors. The ML algorithms were trained on data totaling 46,773 patients with endometrial cancer. As a result, this ML model generated a visual prognosis in the dendrogram format classifying patients into several groups according to five-year survival rates. The authors reported that this ML model improved prognostic prediction for patients with endometrial cancer compared to traditional staging algorithms. The scientists used the EACD ML algorithm for endometrial cancer and demonstrated an improved ability to recognize survival compared to the traditional TNM stage. A number of positive changes in prognostic ability were noted based on the classification of existing pathological variables, as well as the inclusion of other factors in the model, such as age and tumor degree.

## 5. Existing Barriers

Despite the prospects of using ML to solve oncology problems, a number of challenges remain, in particular, the task of data collection and management; the lack of reliable reporting standards; the task of standardizing data; bias inherent in a set of training data; the relative lack of prospective clinical trials. The Existing Minimal Common Oncology Data Elements (mCODE) Initiative is aimed at solving the problem of data collection standardization [67]. To identify signs of patient distress, some institutions are beginning to use a patient-reported outcome measures (PROM) mechanism [68]. Another important problem is the bias of the data on which the models are trained. One of the reasons is the underrepresentation of some population groups (adolescents, women, etc.) during clinical trials, based on which the datasets are formed. The lack of training data for some groups of people can lead to erroneous predictions of the ML model, which is unacceptable for further transfer to clinical practice. It is necessary to ensure a wider inclusion of underrepresented population groups in the training data, which will take a significant amount of time. In addition, the modification of existing solutions will be required, which will mitigate possible systematic errors in datasets [69]. The problem of establishing a standard for reporting on the ongoing research in AI is associated with the high sensitivity of DL methods to source data and difficulties in reproducibility, which may limit the widespread use of AI in oncology. Another problem that has not been fully solved is the insufficient number of studies that compare traditional methods of treating diseases with the methods used with the help of AI. Conducting randomized controlled trials (RCTs) is necessary despite its high cost. Regulatory and legal problems arising with the application of ML methods in medicine have been the subject of discussion for many years. The lack of case law significantly limits the resolution of medical liability issues. In the vast majority of cases, AI implementation in medical practice is used not as a decision-making tool, but as a confirmation of a decision made by a human specialist. Reaching consensus on the regulatory framework for AI-based tools, developing clear standards for evaluating the effectiveness of AI-based disease treatment tools, and conducting new research on the introduction of these tools into medical practice are necessary actions to address the many challenges that limit the potential of using AI in oncology and medicine in general.

## 6. Conclusions

We analyzed 70 papers describing the use of ML methods for the diagnosis of cancer, prediction of the degree of survival, as well as correction and treatment planning. ML is used to deal with various types of cancer, such as carcinoma, glioma, endometrial cancer, prostate cancer, etc. AI finds its application in such modern areas as genomic screening, precision oncology, medicine personalization, and targeted drug delivery. This shows that the progress in the use of AI in medicine is growing rapidly. Various ML models for solving different challenges were considered, such as DT, k-NN, NB, etc., and the results were analyzed. The peculiarities of using various ML models influencing the choice of a specific algorithm for solving each specific task were emphasized. We discussed the prospects of using ML in oncology as a full-fledged tool for the diagnosis and localization of malignant tumors. However, there are a number of barriers that arise when using AI in medicine, related to local, regulatory, and ethical foundations. In this review, we highlighted the importance of solving these issues.

## Figures and Tables

**Figure 1 cancers-16-01100-f001:**
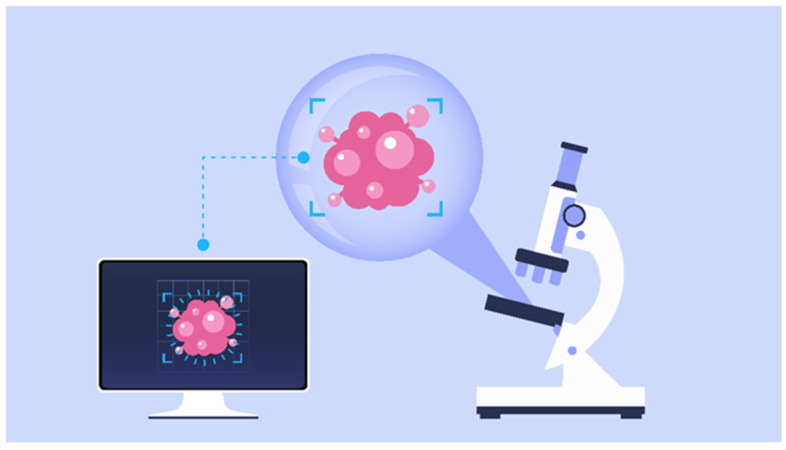
Diagnosis of oncology using computer technology.

**Figure 2 cancers-16-01100-f002:**
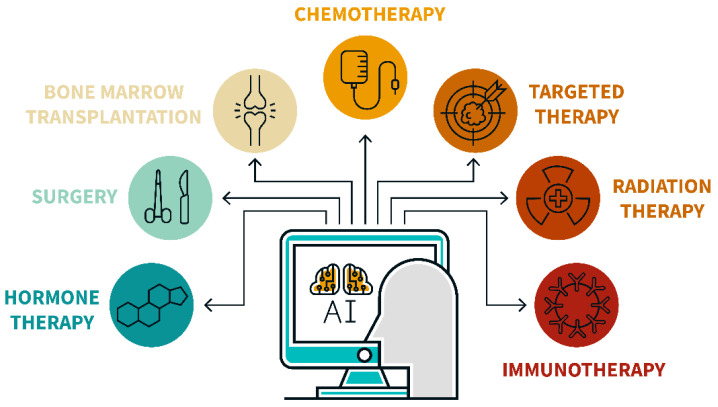
Treatment planning.

**Figure 3 cancers-16-01100-f003:**
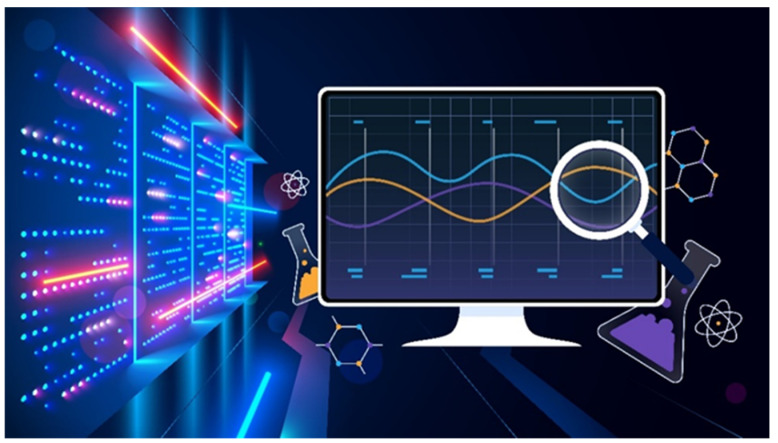
Patient survival prognosis.

## Data Availability

No new data were created or analyzed in this study. Data sharing is not applicable to this article.

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
