# Peer review of "Machine Learning Meets Cancer"

_cancers, 2024, doi:10.3390/cancers16061100_

Round 1

Reviewer 1 Report

Comments and Suggestions for Authors

The paper you're describing offers an overview of the growing role of machine learning (ML) in oncology for various applications, including medical image analysis, treatment planning, patient survival prognosis, and on-site drug synthesis. While the topic is both relevant and timely, there are several minorrevisions that could significantly enhance the quality and impact of the paper:

1.  Ensure that the literature review includes the most recent and relevant studies. The current state of ML in oncology is rapidly evolving, and the paper must reflect the latest advancements and findings.

2. Incorporate studies that compare ML approaches with traditional methods in oncology. This would provide a clearer picture of the advancements and limitations of ML in this field.

3. The paper should delve deeper into the specific ML techniques used in oncology, discussing the pros and cons of various algorithms like deep learning, neural networks, decision trees, etc.

4. Discuss the sources and quality of data used for ML in oncology. Issues like data privacy, standardization, and bias should be addressed.

5. Expand the scope to include a broader range of oncology subfields where ML is being applied, such as genomics, personalized medicine, and immunotherapy.

6. Address the ethical and legal implications of using AI in cancer diagnosis and treatment. This includes patient consent, data security, and the potential for algorithmic bias.

7.Include case studies or examples of real-world applications where ML has made a significant impact in oncology.

8.Clearly outline the current challenges and limitations in the application of ML in oncology. This should cover technical, ethical, and practical aspects.

9. Suggest areas for future research, such as the integration of ML with emerging technologies like CRISPR for cancer treatment.

Comments on the Quality of English Language

Moderate editing of the English language required.

Author Response

Thank you for your review of our paper. We have answered each of your points below.

  1. Ensure that the literature review includes the most recent and relevant studies. The current state of ML in oncology is rapidly evolving, and the paper must reflect the latest advancements and findings.

Response: In this review, we have tried to discuss both proven technologies for using ML in oncology, as well as newer ones that are just beginning to be put into practice. We have added an overview of more recent publications and future trends starting in 2021 (Ref. 1, Ref. 9, Ref. 19 et.al., p. 5, lines 225-236).

  1. Incorporate studies that compare ML approaches with traditional methods in oncology. This would provide a clearer picture of the advancements and limitations of ML in this field.

Response: We have added an overview of studies containing information on the use of machine learning in conjunction with traditional staging methods used in oncology (p.21, lines 943-957).

  1. The paper should delve deeper into the specific ML techniques used in oncology, discussing the pros and cons of various algorithms like deep learning, neural networks, decision trees.

Response: We present an overview of the advantages and disadvantages of various ML models used to solve various problems in oncology. However, the choice of the optimal machine learning model depends on the specific task, the form of representation of the input data and many other factors. So, to solve the problem of improving image descriptions, the random forest algorithm for lung cancer images proved to be the most successful – 98,77. (p. 2, lines 62-85). We present a more detailed discussion of the advantages and disadvantages of using different algorithms in (p. 5, lines 187-210).

  1. Discuss the sources and quality of data used for ML in oncology. Issues like data privacy, standardization, and bias should be addressed.

Response: We have added the discussion of such issues as data standardization and algorithmic bias in (p.20-21, lines 958-988).

  1. Expand the scope to include a broader range of oncology subfields where ML is being applied, such as genomics, personalized medicine, and immunotherapy.

Response: The use of machine learning methods in genomics and precision oncology is rapidly gaining popularity. We have added a discussion of the latest methods in (p.12-14, lines 613-632, 652-678). The use of radiomics as a clinical tool to create more personalized medicine is noted in (p.17, lines 794-800, p.18-19, lines 870-880).

  1. Address the ethical and legal implications of using AI in cancer diagnosis and treatment. This includes patient consent, data security, and the potential for algorithmic bias.

Response: We have considered the legal consequences of using artificial intelligence in the diagnosis and treatment of cancer, including the insufficient provision of some population groups in the source data, which can lead to algorithmic bias in (p.20-21, lines 958-988).

7.Include case studies or examples of real-world applications where ML has made a significant impact in oncology.

Response: Speaking about the application of ML methods in real cases, we note the development of the “SkinVision” application (p.9, lines 410-417), An overview of the endometrial cancer prognostic system developed by A.Praiss et. al. is also provided (p.20, lines 943-957).

8.Clearly outline the current challenges and limitations in the application of ML in oncology. This should cover technical, ethical, and practical aspects.

Response: We have added an overview of the current challenges faced using AI in medicine and in oncology in (p.20-21, lines 958-988). We have noted both ethical and regulatory and legal issues that significantly limit the spread of machine learning for use in clinical practice.

  1. Suggest areas for future research, such as the integration of ML with emerging technologies like CRISPR for cancer treatment.

Response: We have added a discussion of ML usage trends in conjunction with the increasingly popular CRISPR method in oncology (p. 8, lines 352-380).

Reviewer 2 Report

Comments and Suggestions for Authors

I kindly bring the following suggestions to your attention.

1- There is too much repetition of English abbreviations in the article, for example ( CNN ) is explained every time.

2- Some references are misplaced in the sentence (e.g. Line 420) in [28].

3- Tables of the tests performed should be given and mathematical formulas should be given if necessary (Example Anova test results)

4- The Abstract section of the article should be written following scientific rules, including a summary of the work done from the beginning. 

5- The abstract should not contain general information. It should not contain Intro sentences and should contain the motivation and summary of the study.

6-The conclusion is too short. Also, the Conclusion should include what was done in the article and its results. The importance of the results in the literature should be mentioned. This conclusion should be written in accordance with the scientific literature. 

7- Figure 1 and Figure 3 are visually very nice but they don't seem to make any sense. I suggest it be kindly removed.

Comments on the Quality of English Language

I kindly bring the following suggestions to your attention.

1- There is too much repetition of English abbreviations in the article, for example ( CNN ) is explained every time.

2- Some references are misplaced in the sentence (e.g. Line 420) in [28].

3- Tables of the tests performed should be given and mathematical formulas should be given if necessary (Example Anova test results)

4- The Abstract section of the article should be written following scientific rules, including a summary of the work done from the beginning. 

5- The abstract should not contain general information. It should not contain Intro sentences and should contain the motivation and summary of the study.

6-The conclusion is too short. Also, the Conclusion should include what was done in the article and its results. The importance of the results in the literature should be mentioned. This conclusion should be written in accordance with the scientific literature. 

7- Figure 1 and Figure 3 are visually very nice but they don't seem to make any sense. I suggest it be kindly removed.

Author Response

Thank you for your comments. Our answers to your points are as follows.

1.There is too much repetition of English abbreviations in the article, for example ( CNN ) is explained every time.

Response: All repeated explanations of abbreviations have been removed from the text of the manuscript.

  1. Some references are misplaced in the sentence (e.g. Line 420) in [28].

Response: We have moved some links in the text of manuscript for their better perception.

  1. Tables of the tests performed should be given and mathematical formulas should be given if necessary (Example Anova test results)

Response: Some of the studies and papers we mention include test results, correlations, and scale-plots. We would prefer not to reuse these figures in our review, especially since this is due to a request to the authors of these works to use their images, therefore we provide links to the full publications of the authors containing the statistical data and results that we mention.

  1. The Abstract section of the article should be written following scientific rules, including a summary of the work done from the beginning. 

Response: We have added a Simple Summary part (p. 1, lines 14-19).

  1. The abstract should not contain general information. It should not contain Intro sentences and should contain the motivation and summary of the study.

Response: We have fixed the text of the Abstract, removed general information and noted the main motivation of the authors to write this review (p. 1, lines 20-35).

  1. The conclusion is too short. Also, the Conclusion should include what was done in the article and its results. The importance of the results in the literature should be mentioned. This conclusion should be written in accordance with the scientific literature. 

Response: We have fixed the text of the Conclusion (p.21, lines 989-1002).

  1. Figure 1 and Figure 3 are visually very nice but they don't seem to make any sense. I suggest it be kindly removed.

Response: These figures were created specifically for this review, they are unique and cannot be used in other reviews, so we would prefer not to remove them from the manuscript if possible.

Round 2

Reviewer 2 Report

Comments and Suggestions for Authors

This paper is acceptable as edited.